# Targeting Membrane Trafficking as a Strategy for Cancer Treatment

**DOI:** 10.3390/vaccines10050790

**Published:** 2022-05-17

**Authors:** Nydia Tejeda-Muñoz, Kuo-Ching Mei, Pooja Sheladiya, Julia Monka

**Affiliations:** 1Department of Biological Chemistry, David Geffen School of Medicine, University of California, Los Angeles, CA 90095-1662, USA; psheladiya2000@g.ucla.edu (P.S.); jmonka@mednet.ucla.edu (J.M.); 2Division of Pharmacoengineering and Molecular School Pharmaceutics, Eshelman of Pharmacy, University of North Carolina at Chapel Hill, Chapel Hill, NC 27599, USA; kcmei@ad.unc.edu

**Keywords:** macropinocytosis, Wnt signaling, membrane trafficking, V-ATPase, MVBs

## Abstract

Membrane trafficking is emerging as an attractive therapeutic strategy for cancer. Recent reports have found a connection between Wnt signaling, receptor-mediated endocytosis, V-ATPase, lysosomal activity, and macropinocytosis through the canonical Wnt pathway. In macropinocytic cells, a massive internalization of the plasma membrane can lead to the loss of cell-surface cadherins, integrins, and other antigens that mediate cell–cell adhesion, favoring an invasive phenotype. V-ATPase is a key regulator in maintaining proper membrane trafficking, homeostasis, and the earliest developmental decisions in the *Xenopus* vertebrate development model system. Here, we review how the interference of membrane trafficking with membrane trafficking inhibitors might be clinically relevant in humans.

## 1. Membrane Trafficking, Lysosomes, V-ATPase, and Macropinocytosis in the Wnt Pathway

Membrane trafficking, including endocytosis and exocytosis, is very important in the interaction between cells and their environment. Endocytosis mediates the degradation of receptors, hence downregulating signaling pathways [1]. The Wnt pathway is essential for cellular functions, such as cell fate determination, cell migration, cell polarity, neural patterning and organogenesis during embryonic development, including axis formation [2]. The Wnt signaling pathway has been linked to cancer since its discovery. It was found that the overexpression or insertion of *int1*, a mouse gene identical to the Drosophila gene *Wnt1*, in the *Wnt1* region of the genome lead to the formation of tumors [3,4]. Wnt signaling is very complex, belonging to large families of both ligands and receptors. In mammals, there are 19 Wnt ligands and 10 Fzd receptors, in addition to several other pathway activators. Wnt proteins range in length from 350 to 400 amino acids and are post-translationally modified by the O-acyltransferase Porcupine (PORCN), which palmitoylates Wnt proteins in single serine residues. This lipidation forms a binding motif for interacting with Wntless (WLS), which chaperones Wnt proteins to the plasma membrane for secretion. Once secreted, Wnt proteins signal in a paracrine manner, binding nearby receptor complexes [5]. The Wnt ligand binds to the receptors Frizzled and the LDL receptor-related protein 6 (Lrp6), leading to Lrp6 signalosome formation [6]. The Wnt pathway requires endocytosis of a signal receptor for signal transduction to occur [7]. These co-receptors recruit the β-catenin destruction complex containing Glycogen synthase kinase-3 (GSK3). In the absence of Wnt, GSK3 phosphorylates the transcription factor β-catenin, which is subsequently degraded [8] (Figure 1). However, in the presence of Wnt, the ligand binds to the receptors phosphorylating Lrp6, which prevents the complex from localizing into the cytoplasm to mark β-catenin for degradation [6]. Instead, the receptor complex is endocytosed into a vesicle. When activated, the Wnt pathway stabilizes β-catenin, allowing it to localize in the nucleus to interact with other transcriptional regulators, such as TCF/LEF1 (T cell factor/lymphoid enhancer factor family), to trigger the transcription of many different Wnt target genes important for cell fate determination and oncogenesis [9]. So, prior to the activation of β-catenin, the vesicles sequester GSK3, which is found in the same vesicle, into which the receptor complex is sequestered after the binding of the Wnt ligand. As GSK3 and the receptor complex are endocytosed into the vesicles, the endosomal sorting complexes required for transport (ESCRT) machinery move the vesicles into multivesicular bodies (MVB) [10]. As seen in Figure 2, the cell recognizes the decrease in cytoplasmic GSK3 levels, which stabilizes many different Wnt-stabilized-related proteins, such as Ras and PAK1, which trigger macropinocytosis through a pathway called Wnt Stabilization of Proteins (Wnt-STOP) [11,12]. This aberrant activation of the Wnt pathway is strongly implicated in the onset and progression of numerous types of cancer; therefore, this can have therapeutic advantages for cancer treatment, where multiple such targets have been identified with inhibitors acting at different steps of Wnt signaling pathway (Table 1). However, there are currently no FDA-approved specific Wnt-targeting drugs. The reasons for these poor therapeutic benefits are they often lack satisfactory efficacy, specificity, and safety. For instance, due to the crucial roles of Wnt/β-catenin signaling in many cellular functions, many targeted therapies demonstrated obvious side effects. These facts suggest that Wnt/β-catenin signaling-targeted therapies in cancers are still unable to provide a solid clinical translation [13].

Endocytosis is known to play a role in cancer by causing a loss of cell adhesion or morphological polarity, which can lead to the malignant transformation of cells [14,15,16]. For example, the small GTPases Rab proteins, which regulate vesicle transport, protein trafficking, membrane targeting and fusion, also mediate vesicle dynamics, which can work with oncogenic signaling pathways to increase tumor formation [17,18,19,20]. The dysregulated expression of oncogenic Rabs with regard to protein levels or activities, such as Rab1, Rab25, and Rab35, increases proliferation, invasion, and migration through the activation of different signaling pathways. The overexpression of Rab proteins such as Rab3d is seen in breast and lung cancer [21]. Rab2A also facilitates Erk1/2 activation, leading to Zeb1 upregulation and β-catenin nuclear translocation, which promotes tumor initiation. In addition, Wnt signaling is also known to increase endocytosis, which can also implicate a possible connection between Wnt signaling and cancer [22].

## 2. Wnt Signaling Triggers Macropinocytosis 

Recent investigations have shown that the increase in endocytosis in Wnt signaling utilizes macropinocytosis. Pinocytosis (Gr., *pinein*, to drink) is a clathrin-independent endocytic mechanism first described by Warren Lewis (1931). The term macropinocytosis is currently used to designate actin-driven pinocytic vesicles larger than 200 nm. Macropinocytosis is the large nonselective uptake of molecules such as nutrients and other macromolecules in the cellular environment [23]. The macropinosomes that allow the uptake of water and other molecules have membranes derived from the cell plasma membrane’s actin-rich regions, called ruffles, that undergo protrusive movements to allow the vesicle to close and internalize its contents, which are either transported to the lysosome for degradation or to the cell surface [24]. Wnt signaling utilizes macropinocytosis to transport the contents from the cell surface in MVBs to the lysosomes in the cell for degradation [25]. 

Colorectal carcinoma cells (CRC) are known for their increased nuclear β-catenin when APC, a destruction complex protein, is mutated. Furthermore, studies have demonstrated that CRC SW480 cells have robust macropinocytosis which, interestingly, is required for Wnt signaling. This was demonstrated by the decrease in nuclear β-catenin when colorectal cancer cells were treated with macropinocytosis inhibitors such as EIPA [25]. Additionally, mutations in Axin, another component of the destruction complex, increased macropinocytosis [25]. Furthermore, this research shows that macropinocytosis is important for Wnt signaling since nuclear β-catenin accumulation is a marker for an active Wnt signal, which is reduced when cell drinking is inhibited by derivatives of the diuretic amiloride. 

In a different study, by expanding the multivesicular body (MVB) compartment using low doses of the lysosomotropic agent Hydroxychloroquine (HCQ), a strong potentiation of Wnt signaling by LiCl injection in the *Xenopus* embryo was observed, an effect that could be blocked by inhibiting macropinocytosis [26]. Blocking lysosome acidification by V-ATPase via a brief pulse with Bafilomycin A1 (BafA1) at the 32-cell stage inhibited the induction of the primary embryonic axis. The inductive activity of the dorsal determinant Huluwa (Hwa) [27] was also blocked by interfering with lysosome acidification or the MVB-forming ESCRT machinery. These results show that the cell biology of lysosomes plays a fundamental role in vertebrate development; not only linking Wnt signal transduction and membrane trafficking, but also showing that lysosomes/MVBs are required for the activation of the Wnt signal [28].

It has been shown that vacuolar ATPase (V-ATPase), located in the plasma membrane to pump out protons to the extracellular space, is required for macropinocytosis induced by Ras activation [29]. The activity of V-ATPase can be inhibited by BafA1, and similarly results in the inhibition of the macropinocytosis actin machinery. However, BafA1 also inhibits the acidification of intracellular endosomes/MVBs, which are required for sustained macropinocytosis [12,22].

The results portray the physiological relevance of lysosomes in vertebrate development and cancer cell lines [26].

## 3. V-ATPase in Cancer

Vacuolar-ATPase (V-ATPase), initially identified in Saccharomyces cerevisiae and plant vacuoles, is an 830 kDa multi-subunit transmembrane complex. V-ATPases have a similar structure and mechanism of action to mitochondrial F-ATPase (F-type), and several of their subunits evolved from common ancestors. V-ATPase serves to pump protons into the lumen of different endosomal compartments and contribute to endosomal acidification [30] via deuterium discrimination [31] (Figure 3), whereas F-ATPase synthesizes most of the ATP and deuterium-depleted metabolic water in the matrix of mitochondria using an electrochemical proton gradient and oxygen in complex IV [32]. V-ATPase can be found in a variety of locations within cells, including endosomes, lysosomes, Golgi vesicles, secretory vesicles and at the plasma membrane of certain specialized cells [33]. V-ATPase is an attractive target for cancer because it is found to be overexpressed in tumor cell membranes [29]. Furthermore, pH dysregulation is associated with key tumorigenic and metastatic processes, such as migration, invasion, and apoptosis, in addition to the regulation of signaling pathways frequently altered in cancer [34] (Figure 2). In this regard, the decrease in extracellular pH facilitates the degradation of the extracellular matrix (ECM), causing the activation of secreted proteases that degrade the matrix [35]. The importance of pHi (intracellular pH) regulation in cellular transformation has been shown by the ectopic expression of the plasma membrane V-ATPase in fibroblasts. Ectopic V-ATPase expression led to tumorigenic transformation and intracellular alkalinization [36]. The increased invasiveness of breast cancer is correlated with an increased expression of V-ATPase at the plasma membrane of cancerous cells and the upregulation of plasma membrane localization in non-metastatic cells, which increases the metastatic nature of these cells [37,38]. This activity is also related to the epithelial–mesenchymal transition (EMT) of cancer cells. It has been reported that ATP6V1C1 human gene (encoding subunit C) is overexpressed in oral squamous carcinoma cells, and this may promote a greater degree of V1V0 assembly than in normal tissue [39]. Similarly, different subunit genes were found to be overexpressed in drug-resistant cell lines, including ATP6L (ATP6V0C, subunit c) in the case of cisplatin resistance [40]. Remarkably, it was reported that active mTORC1 (mammalian target of rapamycin complex 1) induced the expression of genes encoding several V-ATPase subunits, including isoforms for subunits A, B, C, G, c, and c″, through TEFB transcription factor in both human cells and mice [41].

V-ATPase in the invasive breast cancer MDA-MB-231 cell line was shown to be localized to the plasma membrane, and V-ATPase activity was significantly higher than in less invasive cell lines. Additionally, V-ATPase inhibition induces caspase-dependent apoptosis in invasive tumor cells via mitochondrial pathways [42].

Inhibiting plasma membrane V-ATPase activity in MDA-MB-231 cells with a monoclonal antibody to the V5 epitope of a V5-tagged Voc subunit construct specifically inhibited in vitro invasion [43]. Other studies of breast cancer cells also linked V-ATPase subunit expression to tumor invasion. For example, siRNA knockdown of Voa3 (but not a1, a2 or a4) significantly decreased the invasion of MCF10CA1a cells [38]. Interestingly, in this system, the knockdown of Voa4 led to an increase in Voa3 expression, and a combined knockdown of both Voa3 and a4 led to the greatest reduction in MCF10CA1a cell invasion [37].

The V-ATPase inhibitors, BafA1 and Concanamycin A, interact with the c subunit of V-ATPase and inhibit proton flux. Targeting V-ATPase affects β-catenin nuclear accumulation [26], as it was found that (Figure 3) BafA1 treatment in breast tumor cells decreases metastatic cells [44]. Acidic extracellular pH can also prevent inhibitors from reaching or passing through the plasma membrane of cancer cells [43]. V-ATPase also plays an important role in development [26]. Recently, it was reported that lysosomes play an essential role in the establishment of the initial polarity of the body axis in *Xenopus* embryos, which is controlled by V-ATPase activity. This was demonstrated by blocking lysosome acidification by V-ATPase via a brief pulse with BafA1 at the 32-cell stage, inhibiting the induction of the primary axis. V-ATPase is an important regulator in the Wnt pathway that is involved in several types of cancer [34].

It has been reported that the activated Wnt Lrp6 receptor binds to the peripheral V-ATPase protein ATP6AP2 (also called Prorenin receptor) [45]. ATP6AP2 binds to transmembrane protein 9 (TMEM9), which in turn binds to and activates the V-ATPase that acidifies lysosomes [46]. TMEM9 is a MVB protein that induces secondary axes when overexpressed in *Xenopus* embryos and plays an oncogenic role in colorectal (CRC) and hepatocellular (HCC) carcinomas [47]. Treatment with BafA1 reduces tumor growth in TMEM9-expressing CRC and Hepatocellular carcinoma (HCC) xenografts, demonstrating the importance of lysosomes in tumor progression [47]. 

Wnt, cell adhesion, and membrane trafficking are often active in the same developmental processes, and the crosstalk between them should result in reciprocal regulation. For example, it was reported that the depletion of integrin beta-1 from the plasma membrane by activation of the Wnt pathway promotes cell motility. Importantly, for cancer vaccines, macropinocytosis will change the antigens expressed in the plasma membrane [28]. Knowing how Wnt signaling, macropinocytosis, membrane trafficking, and V-ATPase cooperate will improve our understanding of embryonic development and tumorigenesis. The results highlight that membrane trafficking controlled by the V-ATPase is emerging as a strong therapeutic target to control cancer progression. Therefore, V-ATPase may be a good target for cancer treatment through membrane trafficking [26].

## 4. Targeting Macropinocytosis as a Strategy for Cancer Treatment

Macropinocytosis can be selectively upregulated in different types of cancer. Macropinocytosis can be induced in many cell types upon stimulation with phorbol esters, cytokines and growth factors [48,49] (Figure 4). Therefore, targeting macropinocytosis is a very attractive therapy for cancer. Several studies have shown that 5-(N-ethyl-N-propyl) amiloride (EIPA) or its analog, amiloride, can inhibit both macropinocytosis and actin polymerization by targeting NHE (sodium–hydrogen exchanger) [50]. Similarly, v-ATPase inhibitors, such as BafA1 or Concanamycin A, can also result in the inhibition of macropinocytosis and reduction in intracellular amino acid levels through inhibition of V-ATPase activity. Macropinocytosis can be also blocked by imipramine, a tricyclic antidepressant that blocks membrane ruffle formation in macrophages, dendritic cells, and cancer cells independently of NHE.

Inhibitors of the EGFR inhibitor, such as gefitinib, can suppress the macropinocytosis pathway in NSCLC (non-small cell lung cancer) cells [51]. The Galectin-3 inhibitor GCS-100 disrupts the interaction between Galectin-3 and integrin αvβ3, inhibiting the macropinocytosis pathway in KRAS-addicted lung and pancreatic cancer cells. The DOCK1 (Dedicator of cytokinesis 1) inhibitor 1-(2-(3′-(trifluoromethyl)-[1,1′-biphenyl]-4-yl)-2-oxoethyl)-5-pyrrolidinylsulfonyl-2(1H)-pyridone (TBOPP) can repress DOCK1-mediated macropinocytosis in RAS-transformed cancer cells [52]. Cytochalasin D [53] and PI3K inhibitors (such as Wortmannin and LY294002) can also block macropinocytosis [54]; [55]. The lysosomal inhibitor palmitoyl-protein thioesterase 1 (PPT1), can suppress lysosomal activity, which plays a critical role in degrading proteins during macropinocytosis [56]. mTOR inhibitors have also been reported to suppress the uptake or the processing of lysosomal extracellular protein scavenging [57]. 

Unfortunately, the currently available pharmacological inhibitors of macropinocytosis interrupt other endocytic processes and have nonspecific endocytosis-independent effects. These possible endocytosis-unrelated effects on ion transport, intracellular pH, lysosomal function, and the cytoskeleton limit their use as pharmacological inhibitors of macropinocytosis. Recent evidence supports the view that macropinocytosis is also regulated at the level of GSK3 because the overexpression of the dominant-negative GSK3-GFP (DN-GSK3-GFP) induced macropinocytic TMR-dextran uptake in HCC cells [21]. Glycogen synthase kinase-3 (GSK-3) is a widely expressed serine/threonine kinase that is constitutively active, regulates multiple signaling pathways, and is found at high levels in many diseases such as cancer. Therefore, GSK3 is an attractive target, which involves macropinocytosis, Wnt signaling, membrane trafficking, and cancer, and it can also be a direct regulator of V-ATPase [26]. 

## 5. Conclusions

Macropinocytosis is not the only metabolic pathway in nutrient-poor conditions, such as those found in tumors. For example, there are other metabolic pathways, such as clathrin-mediated endocytosis (CME) and caveolae-mediated endocytosis (CVE), which can also effectively increase nutrient acquisition in cancer cells. Therefore, studies that focus on combining macropinocytosis inhibitors with other metabolic pathway inhibitors may improve the outcomes of cancer therapies. This can facilitate targeted therapeutics without lending systemic toxicity stemming from the broad inhibition of all endocytic pathways. Knowing how Wnt cooperates with V-ATPase, membrane trafficking, and Wnt signaling will help us improve development of efficient drugs for cancer treatment.

## Figures and Tables

**Figure 1 vaccines-10-00790-f001:**
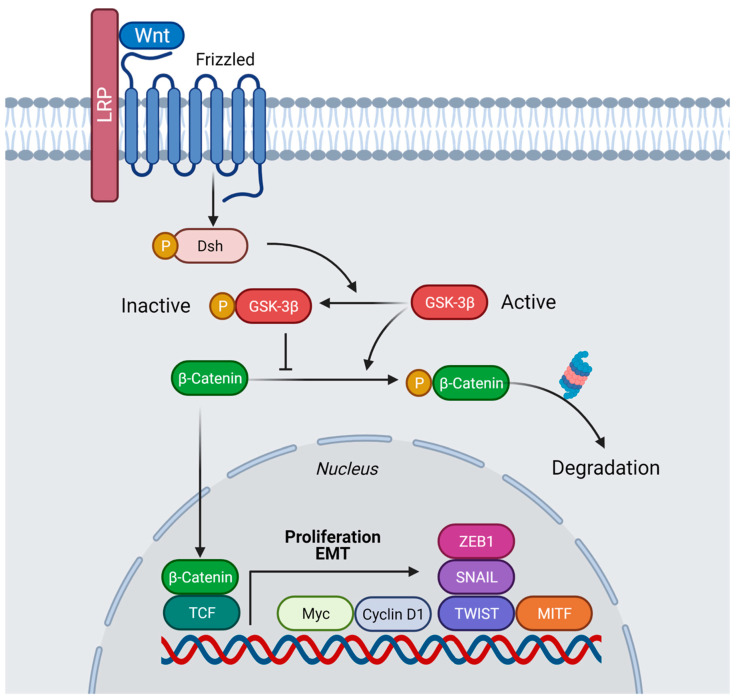
**Model of the Wnt/β-catenin pathway in presence of Wnt ligand.** Binding of Wnt to the receptors Frizzled (Fz) and Lrp6 leads to inhibition of β-catenin degradation. After stabilization, β-catenin is translocated into the nucleus and interacts with members of the TCF/Lef-1 family of transcription factors to co-activate expression of numerous oncogenes involved in proliferation and migration, in particular Cyclin D1 and c-myc, as well as other genes, including Twist, Snail, ZEB1, and MITF, thus, facilitating EMT. Created with BioRender.com (accessed on 29 April 2022).

**Figure 2 vaccines-10-00790-f002:**
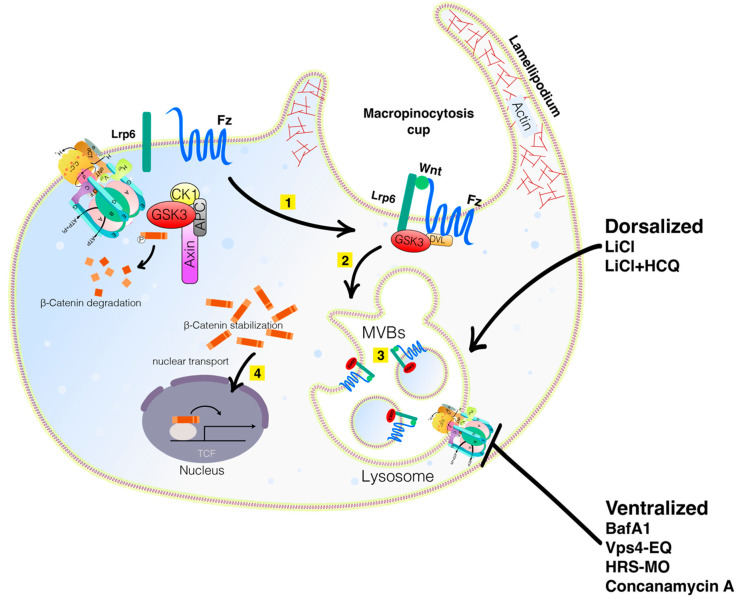
**Wnt signaling involves macropinocytosis, V-ATPase, MVBs, membrane trafficking, and lysosomes.** Sequestration of GSK3 is a vital step in the activation of the canonical Wnt pathway. When the Wnt ligands bind to the Fz receptor and the Lrp6 co-receptors (Step 1 in yellow), GSK3 is translocated into the membrane. It is then internalized into an early endosome and subsequently into MVBs (Step 2). The sequestration of GSK3 and the destruction complex activate the Wnt pathway (Step 3). Lysosomal activity is critical for dorsal development. Mimicking Wnt signaling with LiCl can dorsalize embryos, an effect that is even more pronounced with LiCl plus HCQ. Inhibiting lysosomal activity with BafA1 or Concanamycin A or interfering with the MVB formation with VPS4-EQ or HRS-MO ventralizes embryos. Wnt and cell adhesion are often active in the same processes and crosstalk between them exists by reciprocal regulation and sharing of components. Knowing how Wnt signaling and cell adhesion cooperate will improve our understanding of embryonic development decisions and carcinomas. Diagram based on findings reported in Tejeda—Muñoz et al., 2022, with permission from Proceeding of the National Academy of Science and Creative Commons.

**Figure 3 vaccines-10-00790-f003:**
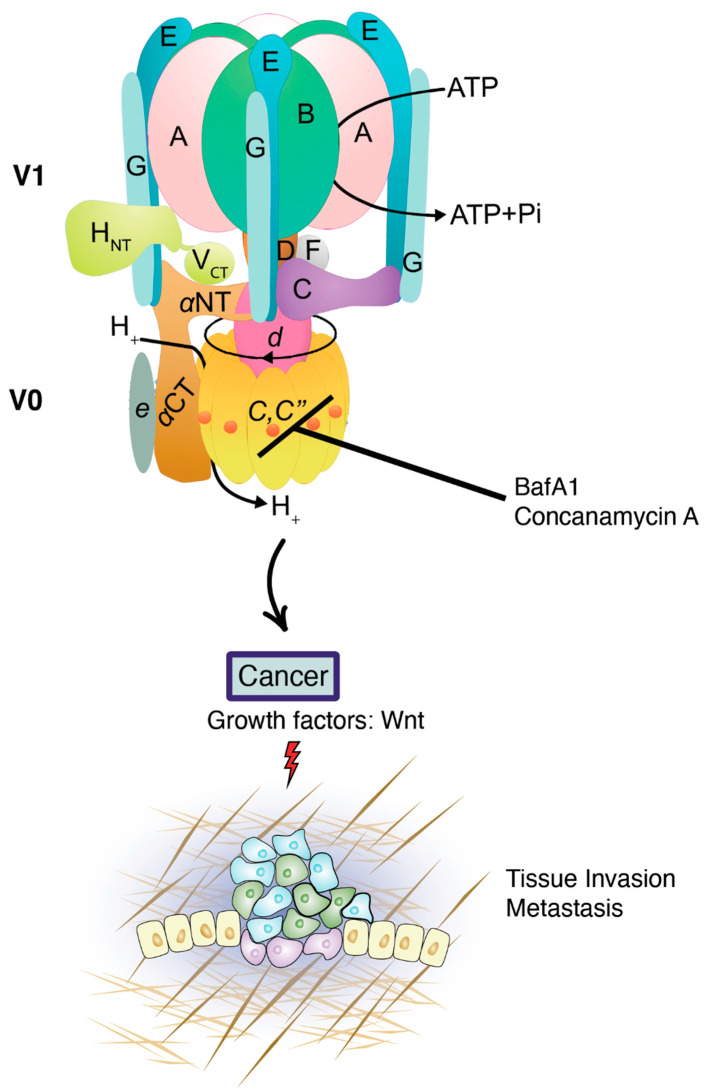
**V-ATPases in cancer.** The V-ATPase is formed of a peripheral V1 domain, which hydrolyzes ATP and an integral V0 domain that translocates protons. The subunits in the catalytic domain (V1) hydrolyze ATP at the cytosolic side of the membrane. The subunits embedded in the membrane form the proton-translocating domain (Vo) that transfers protons from the cytosol to the vesicle lumen. The effects of inhibiting lysosomal function with the specific vacuolar ATPase (v-ATPase) inhibitors such as BafA1 was reported to result in significantly decreased tumor growth, proliferation, and metastasis through the activation of signaling pathways such as Wnt in several types of cancer.

**Figure 4 vaccines-10-00790-f004:**
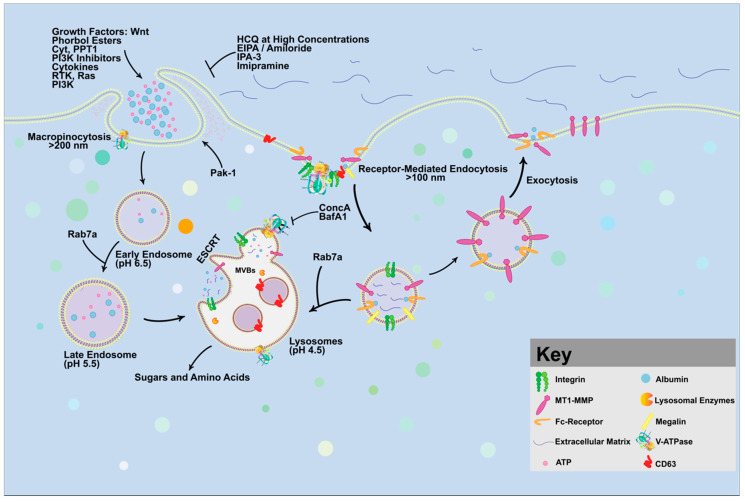
**Targeting macropinocytosis as a promising therapeutic strategy for cancer.** Macropinocytosis is prominent in several types of cancer, such as colon, pancreatic, lung, prostate, and bladder. Through macropinocytosis, serum proteins and a host of extracellular glycoproteins enter the cellular fluid compartment to either be recycled out of the cell or directed to lysosomes for degradation in order to generate key metabolites that fuel cell growth and proliferation. Vacuolar ATPase (V-ATPase) is an essential regulator of RAS-induced micropinocytosis as well as the serine/threonine p21-activating kinases, known as PAK-1 protein. Several inhibitors can block membrane trafficking, macropinocytosis, and lysosomal activity, which affect tumor growth and reveal its important role in cancer. Therefore, macropinocytosis represents a metabolic vulnerability that can be leveraged to therapeutically target macropinocytic tumors by limiting their access to nutrients.

**Table 1 vaccines-10-00790-t001:** Wnt/β-catenin signaling inhibitors.

Wnt/β-catenin Pathway Inhibitors	Name
**Repressor targeting Wnt ligand**	sFRP1 (FRP, SARP2, FrzA) SFRP1, sFRP2 (SARP1) SFRP2 sFRP3 (FrzB, Fritz) FRZB, sFRP4 (FrzB-2) SFRP4, sFRP5 (SARP3) SFRP5 Sizzled, Sizzled2, Crescent, WIF-1, Tiki, Cerberus, Notum, Coco, Dkk-3 (REIC) (DKK3), Soggy (DKKL2), Ipafricept, OMP-18R5, F2.A, IGFBP4, Fz7-21, OTSA-101, Gpr177, Wise, ^90^γ-OTSA-101, OMP-54F28
**Repressor targeting Lrps**	Dkk-1 (DKK1), Sost, Dkk-2 (DKK2), Dkk-4 (DKK4)
**Repressor targeting Fzl**	sFRP1 (FRP, SARP2, FrzA) SFRP1 (inhibits at high concentrations), IGFBP4, OTSA101, OMP-18R5, OMP-54F28
**PORC inhibitors**	WNT974, CGX1321, IWP-2, ETC-159, RXC004, GNF-6231, ^90^γ-OTSA-101, LGK974
**β-catenin/TCF inhibitors**	PFK115-584, CGP049090, CWP291, FL3, ZINC02092166, NC043, iCRT14
**CBP/ β-catenin binding inhibitors**	PRI-724, ICG001, GNE-781, JW67, JW74, NLS-StAx-h, INT-01
**DVL inhibitors**	FJ9, NSC668036, 3289-8625, Niclosamide, J01-017a, sulindac, LM02
**Repressor targeting Axin**	Tankyrase inhibitors; XAV939, IWR-1, NVP-TNKS656, LZZ-02, JW74, WIKI14, K-756, G007-LK, G244-LM, FL3
**β-catenin inhibitors**	COX inhibitors; Aspirin, Celecoxib, Sulindac, 1,25(OH)2D35R, SM08502, PKF115-584, PKF118-310, SAH-BCL9
**Repressor targeting CKI**	Pyrivinium
**Repressor targeting GSK3β**	Genistein
**TCF/LEF inhibitors**	TNIK inhibitor, NCB-0846, PKF115-584, CGP049090
**Repressor targeting DKK**	DKN-01
**Regulates alternative splicing of TCF inhibitors**	SAM68, OMP-54F28

## Data Availability

Not applicable.

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
