# Peer review of "Targeting Membrane Trafficking as a Strategy for Cancer Treatment"

_vaccines, 2022, doi:10.3390/vaccines10050790_

Round 1

Reviewer 1 Report

The review "Targeting membrane trafficking as a strategy for cancer treatment" is a well written basic research review that includes translational aspects for cancer treatment. 

Major remarks:

The activation of the wnt/ß-catenin signaling pathway after wnt-ligand binding to Frizzled and LRP6, subsequent membrane translocation of GSK3, the involvement of pinocytosis, endosomal internalization, MVB´s and the role of vacuolar-ATPase in endosomal acidification, ß-catenin stabilization and nuclear wnt activation should be illustrated more conclusive in figure 1 and 2.

In detail: fig 1: involved genes and growth factors / signaling cascades on proliferation, apoptosis inhibition, migration, tissue invation should be shown (fig 1: nucleus:  cyclinD1, myc, for example); fig 2: numbering of the multi-step process would be heplful.

In addition, wnt-depending transcriptional effects on specific cancer phenotypes should be more clearly described in the text (for example line 119 "metastatic nature of these cells").

An overview of druggable targets and their potential inhibition (e.g. a table of potential wnt ligand antagonists, PORCN inhibitors, Fz antibodies, ß-catenin transcriptional activity inhbitors, v-ATPase inhibitors [concanamycin and bafilomycin as shown in fig 2 and 3] and inhibitors of macropinocytosis) would significantly help to underline the important message of this manuscript using these potential targets for cancer treatment.

Perhaps a list of preclinical / clinical trials of specific inhibitors could be included.

Author Response

Thank you for helping us publish our manuscript entitled “Targeting Membrane Trafficking as a Strategy for Cancer Treatment” in the Vaccines journal.

We have addressed all the reviewers’ comments and suggestions, as indicated in the tracked changes. Their comments greatly improved the paper and we thank them.

The review "Targeting membrane trafficking as a strategy for cancer treatment" is a well written basic research review that includes translational aspects for cancer treatment. 

Major remarks:

The activation of the wnt/ß-catenin signaling pathway after wnt-ligand binding to Frizzled and LRP6, subsequent membrane translocation of GSK3, the involvement of pinocytosis, endosomal internalization, MVB´s and the role of vacuolar-ATPase in endosomal acidification, ß-catenin stabilization and nuclear wnt activation should be illustrated more conclusive in figure 1 and 2.

In detail: fig 1: involved genes and growth factors / signaling cascades on proliferation, apoptosis inhibition, migration, tissue invation should be shown (fig 1: nucleus:  cyclinD1, myc, for example); fig 2: numbering of the multi-step process would be heplful. Thank you for the suggestion. Now, we have included Wnt and EMT target genes in figure 1.

In addition, wnt-depending transcriptional effects on specific cancer phenotypes should be more clearly described in the text (for example line 119 "metastatic nature of these cells"). Now, we have included the type of cancer.

An overview of druggable targets and their potential inhibition (e.g. a table of potential r, PORCN inhibitors, Fz antibodies, ß-catenin transcriptional activity inhbitors, v-ATPase inhibitors [concanamycin and bafilomycin as shown in fig 2 and 3] and inhibitors of macropinocytosis) would significantly help to underline the important message of this manuscript using these potential targets for cancer treatment. Thank you for the suggestion. Now, we have incorporated a table with Wnt inhibitors at the different levels.

Perhaps a list of preclinical / clinical trials of specific inhibitors could be included. Thank you for the comment. Now we have included an explanation about the FDA-approved specific Wnt-targeting drugs.

Reviewer 2 Report

This is potentially an interesting review article that unfortunately lacks critical historic observations regarding the role of selective proton tunneling and pH control via deuterium discrimination, i.e. by its biological fractionation; in various ATPase nanomotors of the mammalian cell. Increased pH and tumorigenicity of fibroblasts expressing a yeast proton pump were described in 1988 (https://doi.org/10.1038/334438a0), for instance, which is an important to trigger mechanisms for increased V-ATPase activity in tumor cells, as argued by the authors in their review. This is due to the fact that deuterons cannot replace protons in active transport processes (https://doi.org/10.1016/0014-5793(90)80248-H) and increases in deuterium content in tumor cells reduce free proton availability, thus elevate pH due to deuterium's low dissociation constant from oxygen in interfacial water compartments of human cells (https://doi.org/10.1186/1742-4682-4-9).

Vacuolar-ATPase plays a primary role in deuterium depletion as a compensatory mechanism for acidification of intracellular organs, including that of the intermembrane space of mitochondria. The authors list a number of cell compartments that utilize the proton pumping function of V-ATPase, yet, mitochondria need to be added. This is important as naturally occurring deuterium (https://doi.org/10.1016/0014-5793(93)81479-J) and its intracellular regulation via continuous water proton exchange reactions in the Krebs-Szent-Györgyi cycle (https://doi.org/10.1016/j.mehy.2015.11.016) are essential for the normal growth rate of cells. 

The authors should include intracellular deuterium depletion (deupletion) that can act as a metabolic therapeutic adjuvant and deupletion can be initiated via diet (https://doi.org/10.1038/s41598-020-62853-8) and V-ATPase activity, alike, among other mechanisms described elsewhere for various cancer types (https://doi.org/10.1093/neuonc/now284), which may also be intended to list in this comprehensive review. The field of deuterium discrimination in metabolomics and biological chemistry is called deutenomics with many applications in translational and clinical medicine, where V-ATPase plays a key role to regulate tumor growth. The authors should be aware of these new developments in biology while also alerting their readers.

Author Response

Thank you for helping us publish our manuscript entitled “Targeting Membrane Trafficking as a Strategy for Cancer Treatment” in the Vaccines journal.

We have addressed all the reviewers’ comments and suggestions, as indicated in the tracked changes. Their comments greatly improved the paper and we thank them.

This is potentially an interesting review article that unfortunately lacks critical historic observations regarding the role of selective proton tunneling and pH control via deuterium discrimination, i.e. by its biological fractionation; in various ATPase nanomotors of the mammalian cell. Increased pH and tumorigenicity of fibroblasts expressing a yeast proton pump were described in 1988 (https://doi.org/10.1038/334438a0), for instance, which is an important to trigger mechanisms for increased V-ATPase activity in tumor cells, as argued by the authors in their review. This is due to the fact that deuterons cannot replace protons in active transport processes (https://doi.org/10.1016/0014-5793(90)80248-H) and increases in deuterium content in tumor cells reduce free proton availability, thus elevate pH due to deuterium's low dissociation constant from oxygen in interfacial water compartments of human cells (https://doi.org/10.1186/1742-4682-4-9). Thank you for the suggestion. Now we have incorporated in the text the work from Perona et al., 1988

Vacuolar-ATPase plays a primary role in deuterium depletion as a compensatory mechanism for acidification of intracellular organs, including that of the intermembrane space of mitochondria. The authors list a number of cell compartments that utilize the proton pumping function of V-ATPase, yet, mitochondria need to be added. This is important as naturally occurring deuterium (https://doi.org/10.1016/0014-5793(93)81479-J) and its intracellular regulation via continuous water proton exchange reactions in the Krebs-Szent-Györgyi cycle (https://doi.org/10.1016/j.mehy.2015.11.016) are essential for the normal growth rate of cells. 

The authors should include intracellular deuterium depletion (deupletion) that can act as a metabolic therapeutic adjuvant and deupletion can be initiated via diet (https://doi.org/10.1038/s41598-020-62853-8) and V-ATPase activity, alike, among other mechanisms described elsewhere for various cancer types (https://doi.org/10.1093/neuonc/now284), which may also be intended to list in this comprehensive review. The field of deuterium discrimination in metabolomics and biological chemistry is called deutenomics with many applications in translational and clinical medicine, where V-ATPase plays a key role to regulate tumor growth. The authors should be aware of these new developments in biology while also alerting their readers. Thank you for the suggestion. Now we have included that the V-ATPases have a similar structure and mechanism of action to mitochondrial F-ATPase (F-type), and several of their subunits evolved from common ancestors. V-ATPase serves to pump protons into the lumen of different endosomal compartments and contribute to the endosomal acidification [30] (Figure 3), whereas F-ATPase synthesizes most of the ATP in mitochondrial using an electrochemical proton gradient (line 124). We want to focus in the V-ATPase, that’s why we didn’t talk deeply about the F-ATPase.

We have also included that V-ATPase is involved in the regulation of mTOR; Remarkably, it has been reported that active mTORC1 (mammalian target of rapamycin complex 1) induces the expression of genes encoding several V-ATPase subunits, including isoforms for subunits A, B, C, G, c and c'', through TEFB transcription factor in both human cells and mice [39] (line 148).

Reviewer 3 Report

Macropinocytosis plays a role in nutrition uptake of cancer, promoted by mutated Ras mediated by Rac1 and regulated by vacuolar ATPase. Other signaling pathways like P13K are involved also. Additionally, internalization pathways are involved in integrin trafficking and can affect the migration and invasion behavior of cells. Therefore, macropinocytosis could be a target for cancer treatment.

In their manuscript titled: “Targeting Membrane Trafficking as a Strategy for Cancer Treatment” the authors gave a well organized and structured overview of triggered macropinocytosis by the Wnt pathway, the role of v-ATPase in cancer and the effects of their inhibitors.

However, as mentioned in chapter 4, pharmacological inhibitors interrupt other endocytosis processes, which limits application by side effects.

The manuscript is well written allowing only minor comments:

Textblock Figure 1: line 127: “etc.” very casually formulation?

Textblock Figure 2: explanation of figure is not complete: the blocks dorsalized / ventralized are not explained, abbreviations of factors should be harmonized (see LRP6 (F1) but Lrp6 (F2))

page 6 line 199: “acronym” is followed by a bracket  

Author Response

Thank you for helping us publish our manuscript entitled “Targeting Membrane Trafficking as a Strategy for Cancer Treatment” in the Vaccines journal.

We have addressed all the reviewers’ comments and suggestions, as indicated in the tracked changes. Their comments greatly improved the paper and we thank them.

In their manuscript titled: “Targeting Membrane Trafficking as a Strategy for Cancer Treatment” the authors gave a well organized and structured overview of triggered macropinocytosis by the Wnt pathway, the role of v-ATPase in cancer and the effects of their inhibitors.

However, as mentioned in chapter 4, pharmacological inhibitors interrupt other endocytosis processes, which limits application by side effects. Thank you, now we have added the limitations of the pharmacological inhibitors.

The manuscript is well written allowing only minor comments:

Textblock Figure 1: line 127: “etc.” very casually formulation?

Thank you for the comment. Now we have removed the etc.

Textblock Figure 2: explanation of figure is not complete: the blocks dorsalized / ventralized are not explained, abbreviations of factors should be harmonized (see LRP6 (F1) but Lrp6 (F2)) Thank you for the suggestion. Now, we have harmonized Lrp6 in the manuscript and explained in figure 2 dorsalized/ventralized concepts.

page 6 line 199: “acronym” is followed by a bracket  Removed

Round 2

Reviewer 2 Report

I thank the authors for adding the Perona 1988 paper about V-ATPase and cell transformation to the manuscript. The revised version also includes a more detailed description that that needs to be modified as "the V-ATPases have a similar structure and mechanism of action to mitochondrial F-ATPase (F-type), and several of their subunits evolved from common ancestors. V-ATPase serves to pump protons selectively into the lumen of different endosomal compartments and contribute to the endosomal acidification [30] via deuterium discrimination (Kotyik et al. https://doi.org/10.1016/0014-5793(90)80248-H) (Figure 3), whereas F-ATPase synthesizes most of the ATP and deuterium depleted metabolic water in the matrix of mitochondria using an electrochemical proton gradient and oxygen in complex IV (https://doi.org/10.1016/j.mehy.2015.11.016).

The above additions are important because, although the authors want to focus on the V-ATPase protein complex, but not F-ATPase, medical deutenomics is a rapidly growing field in medicine with educational credit lectures provided by UC Berkley investigators, among others. The deuterium discriminating role of V-ATPase is an important mechanism in pH control and cell transformation as discussed in the course and elsewhere.

Author Response

Thank you for reviewing our manuscript. We appreciate the detailed assessment and for bringing these points to our attention. I apologize for missing this information in the previous submission. We have performed these changes and have not made any revisions outside of those requested. The new submission includes the following sentence in line 123: “V-ATPases have a similar structure and mechanism of action to mitochondrial F-ATPase (F-type), and several of their subunits evolved from common ancestors. V-ATPase serves to pump protons into the lumen of different endosomal compartments and contribute to the endosomal acidification [30] via deuterium discrimination [31] (Figure 3), whereas F-ATPase synthesizes most of the ATP and deuterium depleted metabolic water in the matrix of mitochondria using an electrochemical proton gradient and oxygen in complex IV [32].”

We greatly appreciate the time you have devoted to our work. Please let us know if there are any additional changes that we should make at the time.